# Safety and Efficacy of Simultaneous Vaccination with Polysaccharide Conjugate Vaccines Against Pneumococcal (13-Valent Vaccine) and Haemophilus Type B Infections in Children with Systemic Juvenile Idiopathic Arthritis: Prospective Cohort Study

**DOI:** 10.3390/vaccines13060644

**Published:** 2025-06-15

**Authors:** Ekaterina Alexeeva, Tatyana Dvoryakovskaya, Dmitry Kudlay, Anna Fetisova, Ivan Kriulin, Elizaveta Krekhova, Anna Kabanova, Vladimir Labinov, Elizaveta Labinova, Mikhail Kostik

**Affiliations:** 1Department of Pediatric Rheumatology, National Medical Research Center of Children’s Health, Moscow 119991, Russia; alekatya@yandex.ru (E.A.); tbzarova@mail.ru (T.D.); 79671819676@yandex.ru (I.K.); anyyyta96@yandex.ru (A.K.); 2Clinical Institute of Children’s Health Named After N.F. Filatov, Department of Pediatrics and Pediatric Rheumatology, I.M. Sechenov First Moscow State Medical University (Sechenov University), Moscow 119991, Russia; 3Department of Pharmacology, Institute of Pharmacy, I.M. Sechenov First Moscow State Medical University (Sechenov University), Moscow 119991, Russia; d624254@gmail.com; 4Laboratory of Personalized Medicine and Molecular Immunology, National Research Center—Institute of Immunology Federal Medical-Biological Agency of Russia, Moscow 115522, Russia; 5Hospital Pediatry, Saint-Petersburg State Pediatric Medical University, Saint-Petersburg 194100, Russia

**Keywords:** systemic juvenile idiopathic arthritis, sJIA, vaccines, pneumococcal vaccine, Haemophilus influenzae type b vaccine, Hib vaccine, biological drugs, disease-modifying antirheumatic drugs, DMARD, macrophage activation syndrome, MAS

## Abstract

**Background**: The introduction of biological drugs into clinical practice for the treatment of children with systemic juvenile idiopathic arthritis (sJIA) allows disease control but increases the risk of infectious events. Infectious events cause immunosuppressive therapy interruptions, leading to disease flare and life-threatening complications, namely macrophage activation syndrome. Our study aimed to evaluate the efficacy and safety of simultaneous vaccination against pneumococcal and Haemophilus influenzae type b (Hib) in children with sJIA. **Methods**: This study included 100 sJIA patients receiving immunosuppressive therapy who were simultaneously vaccinated against pneumococcal and Haemophilus influenzae type b (Hib) infections. The mean age of disease onset was 5.5 years. The median age at vaccination was 10 ± 4.5 years. Clinical and laboratory parameters of sJIA activity, immunization efficacy, and safety, including anti-SP and anti-Hib IgG antibodies, as well as all vaccination-related adverse events (AEs), were recorded in every patient before, 3 weeks after, and 6 months after vaccination. **Results:** At the time of vaccination, 29% of patients did not meet the criteria for the inactive disease stage, as defined by C. Wallace: active joints were present in 34.5% of patients, systemic manifestations (rash and/or fever) were present in 41.3%, and 24.2% of patients had solely inflammatory laboratory activity. The protective titer of anti-SP and anti-Hib IgG antibodies was detected in the majority of patients 3 weeks after vaccination (100% and 93%, respectively). The results remained unchanged (99% and 92%, respectively) for 6 months of follow-up, compared to the baseline (91% and 37%, *p* = 0.000001). Anti-SP IgG and anti-Hib titers raised from 48.3 (18.2; 76.5) and 0.64 (0.3; 3.2) U/mL at the baseline to 103.5 (47.3; 185.4) and 4 (3.5; 4.2) U/mL at D22 and 105 (48.7; 171.8) and 4 (3.8; 4) U/mL (EOS), respectively. Immunosuppressive therapy regimens (combined therapy or biological disease-modifying antirheumatic drug monotherapy) did not influence the immunogenic efficacy of vaccination. The incidence of infectious complications (*p* = 0.0000001) and antibiotic prescriptions (*p* = 0.0000001) decreased by more than two times, to 29.9 and 13.8 events per 100 patient months, respectively, within 6 months after vaccination—the average duration of acute infectious events was reduced by five times after immunization (*p* = 0.0000001). Vaccination did not lead to disease flare: the number of patients with active joints decreased by half compared to the baseline, and the number of patients with systemic manifestations decreased by six times. All vaccine-associated adverse events were considered mild and resolved within 1–2 days. **Conclusions**: Simultaneous vaccination against pneumococcal and Hib infections in sJIA children is an effective and safe tool that reduces the number and duration of infectious events and does not cause disease flare-ups.

## 1. Introduction

Systemic juvenile idiopathic arthritis (sJIA) is the most severe type of JIA, which is characterized by a rapidly progressive course and an adverse prognosis [1]. Biological drugs—inhibitors of IL-1β and IL-6 receptors—approved for the treatment of children with sJIA over two years old [2] according to the modern treatment paradigm of this disease, «treat-to-target,» are highly effective in patients with sJIA [3]; however, they could increase the risk of infectious diseases. According to the REGATE register (France), the frequency of severe infectious complications in patients with rheumatoid arthritis receiving tocilizumab and requiring antibiotic prescriptions and/or hospital admissions was 4.7 per 100 person-years [4]. The BIKER register (Germany), which includes 245 patients with sJIA who received etanercept, IL-1 inhibitors (canakinumab, anakinra), and an IL-6 receptor inhibitor (tocilizumab), reports a high risk of rapidly progressive bacterial infections. Specifically, 110 cases of infectious events were recorded, of which 11 were classified as severe. At the same time, the general infectious rate was 11 and 7 times higher in patients receiving tocilizumab and IL-1 inhibitors compared to those taking etanercept [5].

Infections are the most frequent trigger for disease flare and for a potentially fatal complication of sJIA—macrophage activation syndrome (syn. secondary hemophagocytic syndrome)—even in patients under reasonable sJIA control [6,7].

Streptococcus pneumoniae and Haemophilus influenzae type b (Hib) are both commensals of the human nasopharynx, with the ability to migrate to other niches within the human body, causing various diseases, such as pneumonia, otitis media, and bronchitis [8]. Up to 70% of all cases of pneumonia, about 25% of otitis media, up to 15% of purulent meningitis, and about 3% of endocarditis cases are caused by pneumococcus. The second most important pathogen of bacterial pneumonia is Haemophilus influenzae type b (Hib) [9]. Every year, 1.2 million children over 5 years old die from pneumococcal infection, and up to 200 thousand die from Haemophilus influenza in the world. The mortality rate for pneumococcal pneumonia is 15–30%, and for Haemophilus meningitis, it is 39%. Pneumococcal and Haemophilus infections are among the most common causes of infectious complications in patients receiving immunosuppressants and biological drugs [10].

Vaccination is the most effective method for preventing infection. Standardized vaccination schedules for children and adults are widely used in healthy populations worldwide. However, vaccination coverage of immunocompromised patients currently remains extremely low, mainly due to physicians’ prejudgment, anti-vaccine lobbies, and parents’ fears [11].

Additionally, the immunogenicity and safety of vaccines may vary between healthy individuals and patients with rheumatic diseases who have received immunosuppressive and/or biologic therapy.

Data on the immunogenic efficacy and safety of vaccination in patients with sJIA who are taking conventional immunosuppressants (methotrexate) and biological drugs (TNF, IL-1, and IL-6 inhibitors) are scarce [12].

Our study aimed to evaluate the efficacy and safety of simultaneous vaccination against pneumococcal and Haemophilus infections in children with systemic juvenile idiopathic arthritis.

## 2. Methods

### 2.1. Study Design and Patient Selection

We included 100 sJIA patients in this prospective cohort study, meeting the inclusion and exclusion criteria. The eligible patients were selected voluntarily after 15 years, and their legal representatives were independent of the patient’s age. This study was conducted in the pediatric rheumatology departments of the National Medical Research Center of Children’s Health, Moscow, Russian Federation, and Saint-Petersburg State Pediatric Medical University, Saint Petersburg, Russian Federation, between 2018 and 2020. All patients underwent daily medical observations during the baseline period (Day 0, D0) and three weeks later (Days 1–22, D1–D22). The end-of-study (EOS) observation was conducted six months after the baseline. Day 1 (D1, baseline) was the day of the simultaneous vaccinations. This study’s flow-chart is in Figure 1.

#### 2.1.1. Inclusion Criteria

-The age of patients included in this study is 2–18 years;-Diagnosis of sJIA according to the ILAR (International League of Associations for Rheumatology) criteria;-Treatment with either non-biologic or biologic disease-modifying antirheumatic drugs: tocilizumab, canakinumab, adalimumab, etanercept with or without disease-modifying antirheumatic drugs (DMARDs);-No previous vaccination against pneumococcal and Hemophilus influenza type b infections, except scheduled vaccination against these infections in the first year of life according to the national vaccine schedule. The national vaccine schedule included vaccination against pneumococcal and Hib infections at 3, 4.5, and 6 months, with revaccination after 15 months.

#### 2.1.2. Exclusion Criteria

-Intolerance to vaccine components in the past;-Signs of acute ENT and/or respiratory tract infection;-Mild respiratory, intestinal, or other infections with fever.

### 2.2. Vaccination

We used polysaccharide conjugated vaccines against pneumococcal (vaccine for the prevention of pneumococcal infection polysaccharide, conjugated, adsorbed; “Prevenar 13”; Pfizer; Dublin, Ireland) and Haemophilus influenza type B (vaccine for the prevention of infections caused by Haemophilus influenzae type b; Hiberix; GlaxoSmithKline; Rixensart, Belgium) infections on Day 1 (D1). Both vaccines, each containing 0.5 milliliters, were administered subcutaneously, one after the other, with a 10 min interval. The injections were given in the left deltoid area for the Haemophilus influenza type b vaccine and in the right deltoid area for the pneumococcal vaccine. After the final injection, each patient was closely monitored by the physician for at least half an hour.

### 2.3. The Main Assessments and Outcomes of the Study

The potency of vaccines was evaluated based on their immunogenicity, specifically by measuring the percentage of individuals who developed a protective antibody level 3 weeks and 6 months after receiving the vaccine and/or by observing a doubling of the initial antibody level.

The protective level of antibodies (IgG) for both vaccines was as follows: ≥7 U/mL) for Streptococcus pneumoniae (anti-SP) and ≥1.07 µg/mL (equivalent to U/mL) for Haemophilus influenzae type b (anti-Hib)

### 2.4. Enzyme-Linked Immunosorbent Assay (ELISA)

The participants in this study were sampled for venous blood on the day of their vaccination, as well as three weeks and six months later (with a tolerance of up to three days). The blood samples, in a volume of 5 milliliters, were collected in tubes containing ethylenediaminetetraacetic acid and allowed to sit at room temperature for 30 min in a vertical position. Afterward, the tubes were subjected to centrifugation at 2000 revolutions per minute for a duration of 10 min.

The serum samples were stored in Eppendorf bags (Deltalab S.L., Barcelona, Spain) and kept at −80 °C in an ultracold freezer (ULT Freezer, Haier, Qingdao, China). Prior to the ELISA procedure, the samples were allowed to thaw at room temperature, briefly mixed using a Vortex, and then centrifuged at 2000 rpm for 10 min. The serum samples were diluted 1:101 for the analysis.

The levels of anti-SP antibodies (total IgG titer against serotypes 1–5, 6B, 7F, 8, 9N, 9V, 10A, 11A, 12F, 14, 15B, 17F, 18C, 19A, 19F, 20, 22F, 23F, and 33F) and anti-Hib in serum samples were measured using commercial ELISA kits from Test Line Clinical Diagnostics (Czech Republic) and IBL International Gmbh (Hamburg, Germany), respectively. The ELISA for anti-SP had a lower sensitivity threshold of 2 U/mL, according to the manufacturer’s instructions, and an upper sensitivity threshold of 270 U/mL. For anti-Hib, the lower sensitivity threshold was 0.07 U/mL, and the upper sensitivity threshold was 4 U/mL.

A cutoff value was established for titer values that fell below and exceeded the specified ELISA sensitivity thresholds. The coefficients of variation, as specified by the manufacturers, for the EIA PCP IgG test were 3.8% (intra-assay) and 8.1% (inter-assay) for the Hib IgG ELISA, ≤10% and 9–12%, respectively. The ELISA results were recorded (the intensity of coloration in the wells of a 96-well plate) using an Infinite 200 M plate reader (Tecan, Austria) at a wavelength of 450 nm.

### 2.5. Laboratory Signs of sJIA Activity

Serum concentrations of ESR, high-sensitivity CRP (hs-CRP), and calprotectin (S-100) (MRP8/14 complex) were determined at baseline and 3 weeks after vaccination. The kits hsCRPELISA (Biomerica, Hamburg, Germany) for solid-phase indirect ELISA and MRP8/14S100A8/A9 Calprotectin (BÜHLMANN, Schönenbuch, Switzerland) for sandwich ELISA were used, respectively.

The process of collecting blood samples for analysis and recording the results of ELISA tests was conducted in accordance with the protocol outlined above for evaluating the immunogenicity of vaccines. The cut-off values for high-sensitivity C-reactive protein (maximum value of 8.2 mg/L, analytical sensitivity of 0.1 mg/L) and S-100 protein (maximum value of 2.0 micrograms per milliliter, analytical sensitivity of 0.4 micrograms per milliliter) were employed to assess the level of JIA activity.

### 2.6. Clinical and Laboratory Assessment of sJIA Flare

The impact of vaccination on sJIA activity was assessed by the dynamics of laboratory indicators (CRP, ESR), specifically the difference between baseline values (on the vaccination day) and subsequent values (after 3 weeks), determined in a single blood sample. The dynamics of clinical indicators (systemic manifestations, active joints), physician’s and/or patient’s VAS, CHAQ [13], JADAS71, and sJADAS71 [14] indices were also evaluated 3 weeks after vaccination (compared to the baseline). The remission of sJIA was determined according to the criteria of C. Wallace [15].

### 2.7. The Following Changes Were Additionally Investigated:

(a)Incidence of acute respiratory infection (ARI);(b)Frequency and duration of ARI antibacterial therapy.

The change was evaluated as the difference between indicator values 6 months before and after vaccination for all indicators. These outcomes were recorded according to the face-to-face survey results of parents and sJIA patients on the vaccination day and 6 months after it. Data from medical records were combined with information from parents. ARI episodes were considered cases of increased body temperature (>37 °C) combined with signs of respiratory infection (rhinorrhea, cough, sore throat). ARI complications included cases of ENT infections (otitis, sinusitis, tonsillitis, adenoiditis) and/or lower respiratory tract infections (bronchitis, pneumonia).

### 2.8. Vaccination Safety

Vaccination safety was evaluated by the frequency of adverse event (AE) development, recorded for 30 min (immediate adverse events) and then for 3 weeks after immunization, including the occurrence of intercurrent diseases. AEs were classified as mild and severe. Mild AEs were considered those with local (hyperemia, edema at injection site) and general (rise of body temperature) symptoms, while severe AEs included cases of death, life-threatening events, as well as other conditions requiring prolonged hospitalization (including disease exacerbation) or leading to persistent or significant incapacity or disability.

The physician registered and evaluated adverse events (AEs) occurring immediately after immunization and during the subsequent 3 weeks.

### 2.9. Statistical Analysis

The sample size for the study population, with a margin of error of 5% and a confidence level of 95%, was 80 participants. All statistical analyses were conducted using R software version 4.3.1 (R Foundation for Statistical Computing, Vienna, Austria).

Prior to conducting any statistical analysis, we conducted a preliminary test for the normality of the data distribution using the Shapiro–Wilk test. The majority of the data did not conform to a normal distribution. For numerical data, we presented the median and interquartile range (IQR) or the geometric mean and standard deviation. We also calculated the number of missing values (*n*). For categorical variables, we provided the absolute number (*n*) and the percentage (%) along with the 95% confidence interval (CI). A logarithmic examination of antibody levels was performed before and after vaccination. The changes in antibody titer in each group after vaccination were analyzed using Friedman’s ANOVA and Kendall’s test for three dependent variables and the Wilcoxon matched pairs test for two dependent numerical variables. The proportion of individuals with a protective antibody titer before and after vaccination was compared using McNemar’s test, adjusted for continuity, and separated for each group. Analyses were not possible when one of the proportions equaled 100%. The Bonferroni correction was applied to account for multiple comparisons. Differences are statistically significant at *p*-values of *p* < 0.05 and *p* < 0.017 (0.05/3) after Bonferroni correction for intergroup comparison, with each result being tested multiple times. All values of statistical significance are based on two-tailed tests. Factors that influenced the outcome were examined—the protective titer of anti-SP IgG and anti-Hib IgG were assessed using univariate and multivariate regression analysis.

### 2.10. Ethics

According to the Declaration of Helsinki, written consent has been obtained from all parents and children over 15 years of age. The local Ethics Committee approved the protocol for the trial at the National Medical Research Center of Children’s Health (protocol number 15, dated 2 October 2017).

## 3. Results

### 3.1. Patient Characteristics

This study included 100 patients, with more than half being girls (54%). The mean age of disease onset was 5.5 years. The median age at vaccination was 10 years (4.5). At the time of vaccination, a third of patients (*n* = 29) did not meet the criteria for the inactive disease stage by C. Wallace [15]: active joints were recorded in 34.5% of cases, systemic manifestations presenting as rash and/or fever were recorded in 41.3%, and the remaining patients had solely inflammatory laboratory activity (increased levels of ESR, CRP). Nearly half of the patients (47%) received bDMARD monotherapy, followed by those receiving a combination of bDMARD and nbDMARD (44%) and those receiving nbDMARD monotherapy (9%). Data are in Table 1. Thirty-eight patients (38%) had ENT diseases: tonsillitis (14/100), adenoiditis (12/100), sinusitis (7/100), and otitis (5/100). Thirty-one patients (31%) had respiratory tract infections: laryngitis (5/100), tracheitis (4/100), bronchitis (7/100), and pneumonia (15/100). The frequency of infectious events was 66.8 per 100 patients-months, and the number of antibiotic prescriptions was equal to 44.6 per 100 patients-months 6 months before vaccination.

The IgG anti-SP and anti-Hib antibody levels increased 3 weeks after vaccination and remained stable during the observation period. Almost all patients (100%) had protective antibody levels against pneumococcal infections, and 93% had protective antibody levels against Haemophilus infections by the 22nd day of observation. During the study, 48% and 61% of participants had at least a two-fold increase in baseline levels of antibodies against SP and Hib infection, respectively, at D22 and EOS. There was no significant decrease in the number of patients with protective antibody levels against examined infections by the end of the study. The number of patients with active joints decreased by half compared to the baseline, and the number of patients with systemic manifestations decreased by six times (Table 2).

### 3.2. Analysis of the Influence of Treatment Regimen on the Vaccination Outcome

There were no changes in biological and non-biological disease-modifying antirheumatic drugs in 75 patients for 6 months: 39 patients received combined biological and non-biological DMARD, and 36 received bDMARD monotherapy. Differences in antibody levels against pneumococcus and Haemophilus type b infections have not been found between patients receiving combined therapy and bDMARD monotherapy 3 weeks and 6 months after the immunization. The regressive analysis also did not show any additional effect of non-biological therapy on the persistence of protective antibody levels 3 weeks and 6 months after vaccination (Table 3).

### 3.3. The Frequency of Acute Infectious Events and Antibiotic Use

Almost all patients had at least two episodes of acute infectious events, 67% of which required the prescription of antibacterial therapy within 6 months prior to vaccination. The incidence of infectious complications (*p* = 0.0000001) and antibiotic prescriptions (*p* = 0.0000001) decreased by more than two times, to 29.9 and 13.8 events per 100 patient-months, respectively, within 6 months after vaccination. The average duration of acute infectious events was reduced by five times after immunization (*p* = 0.0000001). Data are in Table 4.

Six months after vaccination, 10/100 patients experienced the following infectious episodes: tonsillitis, 3/100; adenoiditis, 3/100; sinusitis, 1/100; and otitis, 1/100. Nine patients (9%) had respiratory tract infections: laryngitis (3/100), tracheitis (1/100), bronchitis (1/100), and pneumonia (4/100) (Figure 2).

### 3.4. Evaluation of Adverse Events After Vaccination

All patients were under the supervision of a physician in an inpatient rheumatology unit. Adverse events were reported in a quarter of patients (25%) with systemic JIA; all of them were considered to be mild. Local reactions, including pain, hyperemia, and/or edema at the injection site, were reported in 21% of cases and typically resolved within 2–3 days. Fever (body temperature rise ≥ 38 °C) was documented in 4% of cases, which did not require additional pharmacological therapy prescription and resolved independently within 1–2 days. Nobody had any severe AE requiring prolonged hospitalization and/or accompanied by persistent disability.

## 4. Discussion

In our study, the safety and efficacy of simultaneous vaccination against pneumococcal and Haemophilus infections in children with systemic juvenile idiopathic arthritis was confirmed. Some studies are devoted to the safety and efficacy of vaccinating children with rheumatic diseases against the most common infectious agents in the pediatric population, as shown in several systematic reviews [16,17]. However, there is a lack of publications on the subject. The immunogenicity of the 13-valent pneumococcal vaccine in children with systemic juvenile idiopathic arthritis has been studied previously [18], but the efficacy and safety of simultaneous vaccination against pneumococcal and Haemophilus infections in immunocompromised patients, particularly those with sJIA, remain unexplored.

Streptococcus pneumoniae and Haemophilus influenza type b are the most common infectious agents that cause pneumonia, meningitis, and other severe diseases among children. According to the WHO, 735 thousand deaths were the result of pneumococcal infection, while Haemophilus infection led to 363 thousand fatal outcomes before the immunization program’s introduction [19]. Pneumococcal and Haemophilus influenzae type B (Hib) infections are common among immunocompromised patients [20,21]. In addition, a retrospective multicenter study containing 677 patients with JIA demonstrated that infectious complications, also caused by the abovementioned agents, were more frequently reported in patients with systemic JIA. Among all adverse infectious events, 6,5% (*n* = 12) were considered severe and very severe, with seven of them notably occurring in patients with sJIA. It was also shown that the risk of infection development was higher among individuals receiving IL-6 and IL-1 inhibitors [22].

The simultaneous administration of two vaccines reduces the frequency of specific adverse events associated with immunization compared to their separate use, including those caused by pneumococcal and Haemophilus infections [23]. Jan Dolhain et al. confirmed the adequate immunogenicity and safety of diphtheria–tetanus–acellular pertussis–hepatitis B–poliomyelitis–Haemophilus influenzae type b and pneumococcal conjugate vaccine coadministration [24].

There is evidence in the world literature that vaccination with the pneumococcal vaccine reduces the incidence of respiratory infections in general, not only those of pneumococcal etiology [25].

Our study demonstrated the efficacy and safety of simultaneous vaccination against pneumococcal and Haemophilus infections in children with systemic juvenile idiopathic arthritis who were receiving different immunosuppressive drugs. The protective titer of antibodies against pneumococcus was in 91% of patients before vaccination, most likely related to immunization in the first year of life according to the National Vaccination Schedule and/or previous infectious diseases of this etiology. Despite this, there was a two-fold increase in antibody titers in more than half of the patients included in this study that remained at the same level for 6 months after vaccination. High results were also obtained for immunization against Haemophilus infection: the number of patients with protective antibody levels increased by 55% after immunization and remained stable until the end of the observation period. Similar results were observed in our previous study, where 371 children with nonsystemic JIA received simultaneous vaccination against pneumococcal and Haemophilus infections [26].

Generally, this is consistent with available data [27,28]: a sufficient immune response is formed in most patients on immunosuppressive therapy that confirms the vaccination efficacy. However, the issue of long-term resistance to effective immune responses against vaccine-preventable infections remains pertinent. Some studies indicate a gradual decrease in antibody levels after six months or more, necessitating regular monitoring of the antibody titer and revaccination if necessary [29].

It is interesting to study the influence of various biological and non-biological immunosuppressive drugs used in children with rheumatic diseases on the immunogenicity of vaccination. Mariëlle van Aalst et al. demonstrated that patients receiving TNF-α inhibitors as monotherapy had the best immune response to vaccination against pneumococcal infection compared to those prescribed DMARD [30]. Rituximab and abatacept negatively affected the immune response to many vaccines, whereas treatment with IL-6, IL-12/IL-23, and IL-17 antagonists was not associated with a decreased immune response [31]. Our data did not show an adverse effect of non-biological and biological DMARDs on protective antibody levels after 3 weeks, nor did they show resistance 6 months after vaccination. In our previous work, we proved that biological therapy does not affect the synthesis of protective antibodies compared to a control group of conditionally healthy children [32].

The vaccination of children with rheumatic diseases does not lead to disease flares, which was confirmed by a review study involving about 2500 children using vaccines included in the immunization schedules of most countries [17], and is not associated with an increase in JIA activity laboratory markers, such as high-sensitivity C-reactive protein and calprotectin [32]. A comparable safety vaccination profile against Haemophilus infection is also shown in another study of 138 JIA patients [33]. We obtained similar results in the present study: There was no increase in hsCRP and calprotectin levels, as well as in the number of patients with active joints and systemic manifestations (fever, rash, generalized lymphadenopathy, serositis, hepatomegaly, and/or splenomegaly) by the 22nd observation day. Moreover, the number of patients in remission increased to 87% by the end of the study.

The absence of severe adverse events (Aes) throughout the entire observation period indicates good tolerance to simultaneous immunization. All reported vaccine-associated reactions were mild and self-resolved within 1–3 days.

## 5. Conclusions

Simultaneous vaccination against pneumococcal and Haemophilus infections in children with systemic juvenile idiopathic arthritis, regardless of disease stage or immunosuppressive therapy, is an effective and safe tool for preventing infectious complications caused by these pathogens. Immunization does not increase clinical and laboratory sJIA activity markers and is not associated with severe adverse events. The formation and maintenance of protective antibody levels within 6 months after immunization are observed in most patients, despite the use of immunosuppressive drugs, which is accompanied by a reduction in the number and duration of acute infectious events.

## Figures and Tables

**Figure 1 vaccines-13-00644-f001:**
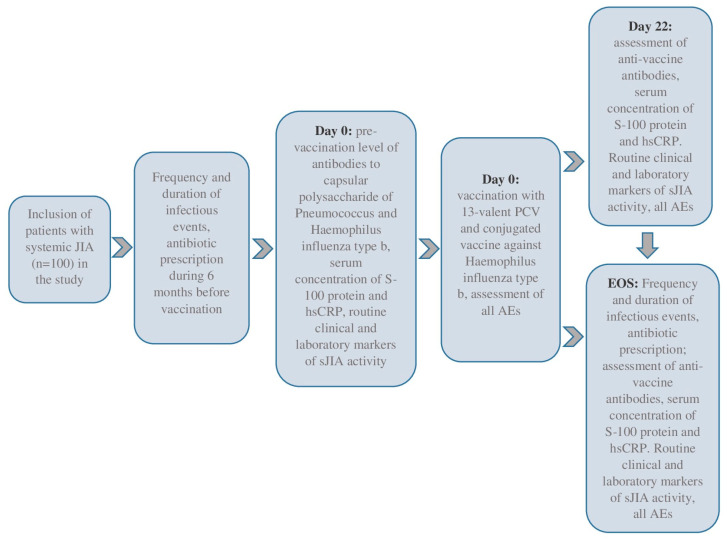
The flow-chart of this study.

**Figure 2 vaccines-13-00644-f002:**
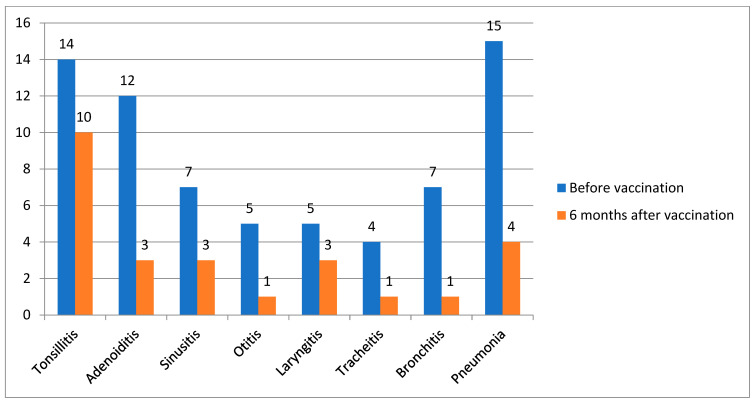
The dynamics of reducing the frequency of infections before and after vaccination, *n* = 100.

**Table 1 vaccines-13-00644-t001:** Patients’ characteristics at baseline.

sJIA Features	Results, *n* = 100 (%)
**Demography**	
Sex, female, *n* (%)	54 (54)
Age at baseline *, years (SD)	10.0 (4.5)
ENT diseases, *n* (%)	38 (38)
**Treatment at baseline**	
NSAIDs, *n* (%)	1 (1)
Glucocorticosteroids, *n* (%)	30 (30)
Methotrexate, *n* (%)	34 (34)
Cyclosporine A, *n* (%)	15 (15)
Mofetyl mycophenolate, *n* (%)	1 (1)
Sulfasalazine, *n* (%)	2 (2)
Colchicine, *n* (%)	4 (4)
**Biologic treatment, *n* (%)**	91 (91)
Tocilizumab monotherapy	24 (24)
Canakinumab monotherapy	10 (10)
Adalimumab monotherapy	1 (1)
Etanercept monotherapy	1 (1)
Tocilizumab + nbDMARD	16 (16)
Canakinumab + nbDMARD	9 (9)
Adalimumab + nbDMARD	1 (1)
Etanercept + nbDMARD	1 (1)
Tocilizumab + GCs	7 (7)
Canakinumab + GCs	4 (4)
Tocilizumab + nbDMARD + GCs	9 (9)
Canakinumab + nbDMARD + GCs	7 (7)
Adalimumab + nbDMARD + GCs	1 (1)

Abbreviations: ENT—ear, nose, and throat; sJIA—systemic juvenile idiopathic arthritis; nbDMARD—non-biological disease-modifying antirheumatic drug; NSAID—non-steroidal anti-inflammatory drug; GCs—glucocorticosteroids; *—calculation made on whole sJIA group.

**Table 2 vaccines-13-00644-t002:** Dynamics of anti-SP and anti-Hib antibodies and the primary sJIA-associated outcomes during the study.

Parameters (*n* = 100) *n* (%)/Me (IQR) [Min; Max]	D1	D22	EOS	*p*1	*p*2	*p*3	*p* Total
Anti-SP IgG titer, U/mL Me (IQR) [min; max]	48.3 (18.2; 76.5) [0.8–299]	103.5 (47.3; 185.4) [5.8–285.2]	105 (48.7; 171.8) [2.03–282.7]	0.0000001	0.0000001	0.249	0.000001
Patients with protective titer anti-SP IgG, *n* (%)	90 (91)	99 (100)	98 (99)	0.008	0.027	1.0	Na
Patients with a two-fold-increased anti-SP titer, *n* (%)	-	48 (48)	60 (61)	0.0000001	0.0000001		
Anti-Hib titer, U/mL, Me (IQR) [min; max]	0.64 (0.3; 3.2) [0.07–101]	4 (3.5; 4.2) [0.2–35]	4 (3.8; 4) [0.2–3.5]	0.0000001	0.0000001	0.423	0.000001
Patients with a protective anti-HIb titer, *n* (%)	37 (37)	92 (93)	91 (92)	0.0001	0.0001	1	Na
Patients with a two-fold-increased anti-HIb titer, *n* (%)	-	61 (62)	78 (79)	0.0000001	0.0000001		
S-100 U/mL (n.v. > 2.9), Me (IQR) [min; max]	3.1 (1.9; 6.9) [0.4–114]	3.2 (2.0; 5.6) [0.4–108]	0.83 (0.5; 1.6) [0.01–114]	0.062	0.0000001	0.0000001	0.000001
hs-CRP, U/mL (n.v. > 8.2), Me (IQR) [min; max]	0.5 (0.2; 1.8) [0.0–40.4]	0.5 (0.2; 1.8) [0.01–102]	0.8 (0.3; 2.1) [0.04–102]	0.291	0.660	0.369	0.087
Patients with systemic features, *n* (%)	12/99 (12.1)	4/99 (4.0)	2/99 (2.0)	0.014	0.017	0.001	na
Patients in remission, *n* (%)	70/99 (70.7)	79/99 (79.8)	84/97 (86.6)	0.039	0.239	0.008	na
Active joints Me (IQR) [min; max]	0.0 (0.0; 0.0) [0.0–13.0]	0.0 (0.0; 0.0) [0.0–13.0]	0.0 (0.0; 0.0) [0.0–8.0]	0.179	0.026	0.066	0.029
Joints with limited ROM Me (IQR) [min; max]	0.0 (0.0; 0.0) [0.0–31.0]	0.0 (0.0; 0.0) [0.0–31.0]	0.0 (0.0; 0.0) [0.0–31.0]	-	0.068	0.084	0.116
Morning stiffness min Me (IQR) [min; max]	0.0 (0.0; 0.0) [0.0–180.0]	0.0 (0.0; 0.0) [0.0–180.0]	0.0 (0.0; 0.0) [0.0–30.0]	0.108	0.012	0.024	0.048
MDVAS, mm Me (IQR) [min; max]	0.0 (0.0; 0.0) [0.0–68.0]	0.0 (0.0; 0.0) [0.0–68.0]	0.0 (0.0; 0.0) [0.0–55.0]	0.043	0.009	0.017	0.017
Patient/parent VAS mm, Me (IQR) [min; max]	0.0 (0.0; 0.0) [0.0–77.0]	0.0 (0.0; 0.0) [0.0–77.0]	0.0 (0.0; 0.0) [0.0–60.0]	0.043	0.005	0.01	0.004
CHAQ, points Me (IQR) [min; max]	0.0 (0.0; 0.0) [0.0–1.25]	0.0 (0.0; 0.0) [0.0–1.25]	0.0 (0.0; 0.0) [0.0–2.0]	0.109	0.075	0.234	0.035
ESR, mm/hr (n.v = 0–20) Me (IQR) [min; max]	2.0 (2.0; 6.5) [0.0–35.0]	2.0 (2.0; 5.0) [0.0–32.0]	2.0 (2.0; 5.0) [0.0–30.0]	0.006	0.005	0.392	0.017
CRP, mg/mL (n.v. < 5) Me (IQR) [min; max]	1.0 (0.8; 1.1) [0.0–242.0]	1.0 (0.5; 1.0) [0.11–287.4]	1.0 (0.2; 1.2) [0.0–102.0]	0.136	0.005	0.033	0.001
JADAS71 points Me (IQR) [min; max]	0.0 (0.0; 0.0) [0.0–30.0]	0.0 (0.0; 0.0) [0.0–28.0]	0.0 (0.0; 0.0) [0.0–16.0]	0.008	0.007	0.026	0.003
JADAS71-CRP points Me (IQR) [min; max]	0.0 (0.0; 0.0) [0.0–38.0]	0.0 (0.0; 0.0) [0.0–28.0]	0.0 (0.0; 0.0) [0.0–16.0]	0.414	0.012	0.010	0.016
sJADAS71	0.00 (0.0; 0.0) [0.0–38.0]	0.00 (0.0; 0.0) [0.0–30.0]	0.0 (0.0; 0.0) [0.0–16.0]	0.647	0.034	0.01	0.020

Footnotes: *p*1—comparison between D1 and D22; *p*2—comparison between D1 and EOS; *p*3 comparison between D22 and EOS. Abbreviations: CHAQ—Childhood Health Assessment Questionnaire; CRP—C-reactive protein; hs-CRP—highly sensitive CRP; ESR—erythrocyte sedimentation rate; Me—median; IQR—interquartile range; ROM—range of motion; VAS—Visual Analogue Scale; MDVAS—physician’s VAS.

**Table 3 vaccines-13-00644-t003:** Predictors of the protective titers of anti-SP IgG and anti-Hib IgG on the 22nd day and end of the study.

Predictor	β	SE	*p*-Value
Protective titer anti-SP IgG titer on D22			
Combined treatment * vs. bDMARD monotherapy	0.002	0.038	0.955
Protective titer anti-SP IgG titer at the EOS			
Combined treatment * vs. bDMARD monotherapy	−0.05	0.037	0.173
Protective titer anti-HIb IgG titer on D22			
Combined treatment * vs. bDMARD monotherapy	0.004	0.05	0.936
Protective titer anti-HIb IgG titer at the EOS			
Combined treatment * vs. bDMARD monotherapy	−0.07	0.07	0.286

Abbreviations: D22—22nd day of the study; EOS—end of the study; bDMARD—biological disease-modifying antirheumatic drug; * Combined treatment means bDMARDs with nbDMARDs.

**Table 4 vaccines-13-00644-t004:** Dynamics of acute respiratory infections before and after simultaneous vaccination.

Infection Burden Indicators, Me (IQR) [Min; Max]	6 Months Prior to Vaccination	6 Months After Vaccination	*p*-Value
Duration of ARI episode, days	10.0 (7.5; 12.0)	2.0 (1.0; 4.5)	0.0000001
The number of ARI episodes per patient	4.0 (3.0; 5.0)	2.0 (1.0; 2.0)	0.0000001
The number of courses of antibacterial drugs	3.0 (2.0; 3.0)	1.0 (0; 1.0)	0.0000001
Patients with ARI, *n* (%)	99 (100)	91 (92)	0.014
Patients require antibacterial drugs, *n* (%)	99 (100)	67 (68)	<0.0001

## Data Availability

The data presented in this study are available on request from the corresponding author.

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
