# Peer review of "Safety and Efficacy of Simultaneous Vaccination with Polysaccharide Conjugate Vaccines Against Pneumococcal (13-Valent Vaccine) and Haemophilus Type B Infections in Children with Systemic Juvenile Idiopathic Arthritis: Prospective Cohort Study"

_vaccines, 2025, doi:10.3390/vaccines13060644_

Round 1
Reviewer 1 Report
Comments and Suggestions for Authors
This prospective cohort study addresses an important clinical question regarding the safety and immunogenicity of simultaneous PCV13 and Hib vaccination in children with sJIA receiving various immunosuppressive therapies. The findings suggest that the vaccination strategy is safe, immunogenic, and potentially reduces infection burden in this vulnerable population. However, several aspects require clarification and refinement to strengthen the conclusions.
Major Comments:
- The manuscript utilizes an ELISA measuring total IgG against a broad pool of 24 S. pneumoniae serotypes (Line 147-148), whereas the PCV13 vaccine targets only 13 specific serotypes. This mismatch is a significant limitation. The observed antibody increases, particularly from a high baseline seropositivity (91%, Line 326), may not solely reflect the response to vaccine serotypes. Increases could be driven by antibodies to non-vaccine serotypes included in the assay but not in the vaccine. The authors should explicitly discuss this limitation and its potential impact on interpreting the vaccine-specific immunogenicity. Ideally, results from a serotype-specific assay (e.g., WHO standard ELISA or OPA) would provide more definitive evidence of vaccine response. If such data are unavailable, caution is needed in claiming vaccine efficacy based solely on this assay.
2.Given that 91% of patients had protective anti-SP IgG titers at baseline, the primary outcome measure (proportion achieving protective titer) is less informative for PCV13 than the fold-increase analysis. The study essentially measures a booster response in the majority of the cohort for PCV13. Stratify the analysis for both PCV13 and Hib based on baseline serostatus (seronegative vs. seropositive). This would allow for a clearer assessment of primary responses (in the seronegative group) versus booster responses (in the seropositive group) and how these might differ, particularly under immunosuppression. Provide more details on the national infant PCV immunization schedule relevant to this cohort to contextualize the high baseline SP titers.
3.The study reports a significant reduction in "acute infectious events" and antibiotic use post-vaccination (Section 3.3, Table 4). This is a key clinical finding. Provide a clearer definition of how "acute infectious events" were identified and recorded prospectively. Crucially, specify the types of infections recorded (e.g., upper respiratory, lower respiratory/pneumonia, otitis media, invasive bacterial disease). Attributing the reduction specifically to protection against S. pneumoniae and Hib requires demonstrating a decrease primarily in infections typically caused by these pathogens.
Minor Comments:
1.Justification or references should be provided for the specific antibody titer cut-offs used to define protection (anti-SP ≥ 7 U/ml, anti-Hib ≥ 1.07 µg/ml - Lines 134-135). Are these based on WHO standards, kit manufacturer recommendations, or specific studies in immunocompromised pediatric populations? Also, clarify the basis for the µg/ml to U/ml equivalence for anti-Hib.
2.Statistical Reporting (Regression Analysis): In Table 3 (regression analysis), please include 95% confidence intervals for the Beta coefficients (β) in addition to p-values. This provides better insight into the precision of the estimates, especially given the non-significant findings for the effect of combined therapy.
Author Response
This prospective cohort study addresses an important clinical question regarding the safety and immunogenicity of simultaneous PCV13 and Hib vaccination in children with sJIA receiving various immunosuppressive therapies. The findings suggest that the vaccination strategy is safe, immunogenic, and potentially reduces infection burden in this vulnerable population. However, several aspects require clarification and refinement to strengthen the conclusions.
Reply: Dear Reviewer! We sincerely appreciate your positive feedback and thoughtful review. Our answers (A) on your queries (Q) are below and highlighted by color in the manuscript.
Major Comments:
Q1. The manuscript utilizes an ELISA measuring total IgG against a broad pool of 24 S. pneumoniae serotypes (Line 147-148), whereas the PCV13 vaccine targets only 13 specific serotypes. This mismatch is a significant limitation. The observed antibody increases, particularly from a high baseline seropositivity (91%, Line 326), may not solely reflect the response to vaccine serotypes. Increases could be driven by antibodies to non-vaccine serotypes included in the assay but not in the vaccine. The authors should explicitly discuss this limitation and its potential impact on interpreting the vaccine-specific immunogenicity. Ideally, results from a serotype-specific assay (e.g., WHO standard ELISA or OPA) would provide more definitive evidence of vaccine response. If such data are unavailable, caution is needed in claiming vaccine efficacy based solely on this assay.
A1. Dear Reviewer! Thank you so much for your comments. The detection of antibodies against 24 antigens could explain the relatively high rate of the patients with positive titer against Pneumococcus at the baseline. This point has been added to the limitation section, but we are absolutely sure that the raise of titers after the vaccination was almost associated with the vaccination. Unfortunately, we had no data of the serotype-specific assay.
Q2. Given that 91% of patients had protective anti-SP IgG titers at baseline, the primary outcome measure (proportion achieving protective titer) is less informative for PCV13 than the fold-increase analysis. The study essentially measures a booster response in the majority of the cohort for PCV13. Stratify the analysis for both PCV13 and Hib based on baseline serostatus (seronegative vs. seropositive). This would allow for a clearer assessment of primary responses (in the seronegative group) versus booster responses (in the seropositive group) and how these might differ, particularly under immunosuppression. Provide more details on the national infant PCV immunization schedule relevant to this cohort to contextualize the high baseline SP titers.
A2. Dear Reviewer! In the Methods point 2.3 we mentioned that two-fold increased analysis was our primary outcome. “The effectiveness of vaccines was assessed by their immunogenicity, namely by the proportion of patients in whom a protective antibody titer was determined 3 weeks and 6 months after vaccination and/or in combination with an increase in titer ≥ 2 times from the initial one.”
The statement “During the study 48% and 61% had at least two-fold increase of the baseline levels of antibodies against SP and 62% and 79% against the Hib infection at D22 and EOS, respectively” has been added to the Results. Also 100% and 99% achieved the protective level of antibodies against SP and 93% and 92% against Hib infection at D22 and EOS. According the seronegative group it is difficult to provide unbiased analysis due to vey low (n=9) patients who had no the protective levels of the antibodies against SP. National vaccine schedule included vaccinations against Pneumococcus and Hib infections at 2-3; 4,5; 6 months vaccination and 15 months revaccination.
Q3. The study reports a significant reduction in "acute infectious events" and antibiotic use post-vaccination (Section 3.3, Table 4). This is a key clinical finding. Provide a clearer definition of how "acute infectious events" were identified and recorded prospectively. Crucially, specify the types of infections recorded (e.g., upper respiratory, lower respiratory/pneumonia, otitis media, invasive bacterial disease). Attributing the reduction specifically to protection against S. pneumoniae and Hib requires demonstrating a decrease primarily in infections typically caused by these pathogens.
A3. Dear Reviewer! This point was assessed prospectively and retrospectively. At the baseline the data about the infections were collected asking the parents and assessment the available medical records and after the baseline the parents were asked to record the information about the infections and again, they were asked at the end of the study about the infections and information about all relevant episodes, its duration and antibiotics treatment were extracted from the available medical documents.
The data about infections, preceding the baseline is provided in the beginning of the Results section (lines 249-254) “Thirty-eight patients (38%) had ENT diseases: tonsillitis (14/100), adenoiditis (12/100), sinusitis (7/100), and otitis (5/100). Thirty-one patients (31%) had respiratory tract infection: laryngitis (5/100), tracheitis (4/100), bronchitis (7/100), pneumonia (15/100). The frequency of infectious events was 66.8 per 100 patients-months, and the number of antibiotic prescriptions was equal to 44.6 per 100 patients-months 6 months before vaccination.”
Six months after vaccination, 10/100 patients had the following infectious episodes: tonsillitis, 3/100 – adenoiditis, 3/100 – sinusitis, 1/100 – otitis. Nine patients (9%) had respiratory tract infection: laryngitis (3/100), tracheitis (1/100), bronchitis (1/100), pneumonia (4/100) (fig. 3).
All cases of the infections were treated at home at out-patients departments and primary care physicians did not provide more detailed information about the type and etiology of the upper respiratory infections. The bacteriological assessment usually is not included in the routine out-patient clinical practice. It obligatory performs only in the cases of hospital admission. In our study there were no hospital admissions due to infectious episodes during the study and six months before the baseline. So, we cannot provide the data about Sp- or Hib-specific efficacy of the vaccines, in the respect of the pathogens. But, in the literature there are the data about the ability of Pneumococcal vaccine reduce the incidence or respiratory infection of different etiology, not only SP-specific. The statement “There is evidence in the world literature that vaccination with pneumococcal vaccine reduces the incidence of respiratory infections in general, not only of pneumococcal etiology [25]” has added in the discussion section.
Minor Comments:
Q1. Justification or references should be provided for the specific antibody titer cut-offs used to define protection (anti-SP ≥ 7 U/ml, anti-Hib ≥ 1.07 µg/ml - Lines 134-135). Are these based on WHO standards, kit manufacturer recommendations, or specific studies in immunocompromised pediatric populations? Also, clarify the basis for the µg/ml to U/ml equivalence for anti-Hib.
A1. Dear Reviewer! The cut-offs were provided by the manufacturer of the kits. The anti-Hib antibodies were measured in U/ml, the correction has done in the manuscript.
Q2. Statistical Reporting (Regression Analysis): In Table 3 (regression analysis), please include 95% confidence intervals for the Beta coefficients (β) in addition to p-values. This provides better insight into the precision of the estimates, especially given the non-significant findings for the effect of combined therapy.
A2. Dear Reviewer! Our statistical package did not provide the 95% CI for Beta coefficients, unfortunately
Dear Reviewer!
Thank you so much for your help and efforts.
I hope the manuscript has become better after your suggestions.
On behalf of the Authors
Mikhail Kostik, MD, PhD, Professor
Reviewer 2 Report
Comments and Suggestions for Authors
I appreciate very much your prospective Cohort study .
Could you make clear that day 1 was Six months before the vaccination day and precise the methodology used to get accurate data before the vaccination.The section 2.7 on Changes could describe the time frame of this longitudinal study.
In the discussion, you could discuss this approach versus a comparison with a placebo cohort .
Also in the discussion section, you may discuss the long term impact of both vaccination and need for boosters.
Author Response
I appreciate very much your prospective Cohort study.
Reply: Dear Reviewer! We sincerely appreciate your positive feedback and thoughtful review. Our answers (A) on your queries (Q) are below and highlighted by color in the manuscript.
Q1. Could you make clear that day 1 was Six months before the vaccination day and precise the methodology used to get accurate data before the vaccination.
A1. Dear Reviewer! The Day 1 (Baseline) it was a day of the simultaneous vaccination against Hid and Sp. The additional clarification added in the manuscript, and the study’s flow-chart added.
Q2. The section 2.7 on Changes could describe the time frame of this longitudinal study.
A2. Dear Reviewer! The study’s flow-chart has added (fig. 2)
Q3. In the discussion, you could discuss this approach versus a comparison with a placebo cohort. Also in the discussion section, you may discuss the long-term impact of both vaccination and need for boosters.
A3. Dear Reviewer! This study did not have a placebo comparison cohort, but in the Discussion (lines 366-368) we mentioned our previous study with the statement: “In our previous work, we proved that biological therapy does not affect the synthesis of protective antibodies compared to a control group of conditionally healthy children [32].” Currently we are doing the second part of this study with long-term follow-up observation and assessment.
Dear Reviewer!
Thank you so much for your help and efforts.
I hope the manuscript has become better after your suggestions.
On behalf of the Authors
Mikhail Kostik, MD, PhD, Professor
Reviewer 3 Report
Comments and Suggestions for Authors
This study presents a prospective cohort study evaluating the safety and immunogenicity of simultaneous pneumococcal and Haemophilus influenzae type b vaccination in children with systemic juvenile idiopathic arthritis (sJIA) receiving immunosuppressive therapy. The topic is timely and clinically relevant, particularly given the increased risk of infections in immunocompromised pediatric patients. While the study is well-conducted and offers useful data, several major issues need to be addressed before the manuscript can be considered for publication.
Major Comments
Lines 27–28, 98–107: The absence of a healthy control group or a disease control group limits the strength of the conclusions regarding vaccine safety and efficacy. A matched cohort would enhance the interpretability of the findings. Please clarify why a control group was not included and discuss its potential impact in the limitations section.
Lines 33–34, 230–233: The paper reports that 29% of patients were not in inactive disease stage, yet 41.3% had systemic manifestations and 34.5% had active joints, suggesting possible overlap or inconsistency. Please clarify how disease activity was categorized and ensure that the percentages are non-overlapping.
Lines 370–374, 338–340: The follow-up duration is 6 months, which may not be sufficient to assess long-term vaccine efficacy, especially in immunocompromised patients. Please address this limitation more explicitly and cite relevant literature about long-term antibody persistence in similar populations.
Minor Comments
Lines 22, 45, 37, 252: Several grammatical errors and awkward phrases are present. A thorough English language review is recommended to improve clarity and readability.
Lines 218–219, 255–260: Some p-values are reported as “0.0000001” which is statistically imprecise. Also, the order of p-values (p1, p2, p3) is confusing without table legends. Use standard reporting format and clarify comparisons in table footnotes.
Lines 123–124: The names and manufacturers of vaccines should be written in full, including country of origin and trade name.
Lines 379–474: Some DOIs are incomplete or repeated, and references appear inconsistently formatted. Ensure all references conform to the MDPI Vaccines reference style and include proper DOIs or PubMed IDs.
No visual summary is included. Consider adding at least one figure to visually depict key results such as IgG titer changes or infection rate reductions.
Comments on the Quality of English LanguageThe manuscript is generally understandable but contains multiple grammatical and syntactic errors. A thorough revision by a native English speaker or professional editing service is recommended.
Author Response
This study presents a prospective cohort study evaluating the safety and immunogenicity of simultaneous pneumococcal and Haemophilus influenzae type b vaccination in children with systemic juvenile idiopathic arthritis (sJIA) receiving immunosuppressive therapy. The topic is timely and clinically relevant, particularly given the increased risk of infections in immunocompromised pediatric patients. While the study is well-conducted and offers useful data, several major issues need to be addressed before the manuscript can be considered for publication.
Reply: Dear Reviewer! We sincerely appreciate your positive feedback and thoughtful review. Our answers (A) on your queries (Q) are below and highlighted by color in the manuscript.
Major Comments
Q1. Lines 27–28, 98–107: The absence of a healthy control group or a disease control group limits the strength of the conclusions regarding vaccine safety and efficacy. A matched cohort would enhance the interpretability of the findings. Please clarify why a control group was not included and discuss its potential impact in the limitations section.
A1. Dear Reviewer! We are absolutely agree with you. This is a weak part of our study and it has mentioned in the Limitations section. Some clinical data about controls from our previous study mentioned in the discussion section Discussion (lines 366-368) “In our previous work, we proved that biological therapy does not affect the synthesis of protective antibodies compared to a control group of conditionally healthy children [32].”
Q2. Lines 33–34, 230–233: The paper reports that 29% of patients were not in inactive disease stage, yet 41.3% had systemic manifestations and 34.5% had active joints, suggesting possible overlap or inconsistency. Please clarify how disease activity was categorized and ensure that the percentages are non-overlapping.
A2. Dear Reviewer! Some patients with active joints also had systemic features and some patients had no systemic features due to the treatment, but had persistent arthritis. So, some patients were overlapped without mistakes.
Q3. Lines 370–374, 338–340: The follow-up duration is 6 months, which may not be sufficient to assess long-term vaccine efficacy, especially in immunocompromised patients. Please address this limitation more explicitly and cite relevant literature about long-term antibody persistence in similar populations.
A3. Dear Reviewer! We are agree that this follow-up period is too short. It has been mentioned in the Limitations according to your suggestions. The second part of this study with long-term follow-up observation and assessment is ongoing now.
Minor Comments
Q4. Lines 22, 45, 37, 252: Several grammatical errors and awkward phrases are present. A thorough English language review is recommended to improve clarity and readability.
A4. Dear Reviewer! The professional English editing done
Q5. Lines 218–219, 255–260: Some p-values are reported as “0.0000001” which is statistically imprecise. Also, the order of p-values (p1, p2, p3) is confusing without table legends. Use standard reporting format and clarify comparisons in table footnotes.
A5. Dear Reviewer! p1 – this is a comparison between D1 and D22; Ñ€2 – this is a comparison between D1 and EOS; and p3 – this is a comparison between D22 and EOS. The information is in the Table’s footnote. I do not understand your comment about “statistically imprecise”. All calculations of p-value are correct.
Q6. Lines 123–124: The names and manufacturers of vaccines should be written in full, including country of origin and trade name.
A6. Dear Reviewer! This information added.
Q7. Lines 379–474: Some DOIs are incomplete or repeated, and references appear inconsistently formatted. Ensure all references conform to the MDPI Vaccines reference style and include proper DOIs or PubMed IDs.
A7. Dear Reviewer! Thank you so much! The reference list has been updated.
Q8. No visual summary is included. Consider adding at least one figure to visually depict key results such as IgG titer changes or infection rate reductions.
A8. The graphical abstract (fig. 1) created and placed after the Abstract. If you want, we can place it at the end of the Results section or anywhere you recommend.
Q9. Comments on the Quality of English Language. The manuscript is generally understandable but contains multiple grammatical and syntactic errors. A thorough revision by a native English speaker or professional editing service is recommended.
A9. Dear Reviewer! The professional English editing done.
Dear Reviewer!
Thank you so much for your help and efforts.
I hope the manuscript has become better after your suggestions.
On behalf of the Authors
Mikhail Kostik, MD, PhD, Professor
Round 2
Reviewer 3 Report
Comments and Suggestions for Authors
The authors have adequately addressed all major and minor concerns raised during the previous review. The manuscript has undergone substantial improvements in terms of clarity, scientific rigor, and formatting.